

# Mesozooplankton grazing minimally impacts phytoplankton abundance during spring in the western North Atlantic

Francoise Morison[1], James Joseph Pierson[2], Andreas Oikonomou[1,3] and Susanne Menden-Deuer[1]

[1] Graduate School of Oceanography, University of Rhode Island, Narragansett, RI, United States of America
[2] Center for Environmental Science, University of Maryland, Cambridge, MD, USA
[3] Institute of Balances of internal water, Hellenic Centre for Marine Research, Athens, Greece

## ABSTRACT

The impacts of grazing by meso- and microzooplankton on phytoplankton primary production (PP) was investigated in the surface layer of the western North Atlantic during spring. Shipboard experiments were performed on a latitudinal transect at three stations that differed in mixed layer depth, temperature, and mesozooplankton taxonomic composition. The mesozooplankton community was numerically dominated by *Calanus finmarchicus* at the northern and central station, with *Calanus hyperboreus* also present at the northern station. The southern station was >10 °C warmer than the other stations and had the most diverse mesozooplankton assemblage, dominated by small copepods including *Paracalanus* spp. Microzooplankton grazing was detected only at the northern station, where it removed 97% of PP. Estimated clearance rates by *C. hyperboreus* and *C. finmarchicus* suggested that at in-situ abundance these mesozooplankton were not likely to have a major impact on phytoplankton abundance, unless locally aggregated. Although mesozooplankton grazing impact on total phytoplankton was minimal, these grazers completely removed the numerically scarce > 10 μm particles, altering the particle-size spectrum. At the southern station, grazing by the whole mesozooplankton assemblage resulted in a removal of 14% of PP, and its effect on net phytoplankton growth rate was similar irrespective of ambient light. In contrast, reduction in light availability had an approximately 3-fold greater impact on net phytoplankton growth rate than mesozooplankton grazing pressure. The low mesozooplankton grazing impact across stations suggests limited mesozooplankton-mediated vertical export of phytoplankton production. The constraints provided here on trophic transfer, as well as quantitative estimates of the relative contribution of light and grazer controls of PP and of grazer-induced shifts in particle size spectra, illuminate food web dynamics and aid in parameterizing modeling-frameworks assessing global elemental fluxes and carbon export.

Corresponding author
Francoise Morison, fmorison@uri.edu

## INTRODUCTION

Zooplankton occupy a pivotal position in pelagic food webs. Grazing by zooplankton represents the major fate of marine phytoplankton production (*Banse, 2013*; *Steinberg & Landry, 2017*), influencing energy transfer to higher trophic levels, nutrient cycling, and carbon flow. As important herbivores, zooplankton act as key mediators of the biological pump that contributes to the ocean drawdown of carbon from the atmosphere (*Ducklow, Steinberg & Buesseler, 2001*). In the North Atlantic, grazing by zooplankton has been identified as an important factor influencing the formation, timing, and magnitude of the spring bloom (*Riley, 1946*; *Cushing, 1959*; *Evans & Parslow, 1985*; *Banse, 1994*; *Behrenfeld & Boss, 2014*). Empirical quantification of herbivorous zooplankton feeding rates is therefore crucial to understanding phytoplankton blooms, trophic linkages, and biogeochemical cycles.

Among the extremely diverse zooplankton assemblage, the micro- (20–200 µm) and meso- (200 µm–20 mm) size classes (*Sieburth, Smetacek & Lenz, 1978*) are recognized as the major grazers of phytoplankton (*Steinberg & Landry, 2017*). Microzooplankton are predominantly represented by herbivorous protists with rapid doubling rates on the order of one or a few days, many of which function as mixotrophs (*Stoecker et al., 2017*). Mesozooplankton are a functionally diverse group of metazoan grazers that is largely composed of copepods (*Turner, 2004*) and occupies multiple trophic levels in planktonic food webs (*Calbet & Saiz, 2005*; *Saiz & Calbet, 2011*). Compared to microzooplankton, mesozooplankton have generation times on the order of weeks and months and are numerically less abundant (*Schmoker, Hernández-León & Calbet, 2013*). Thus, due to their capacity to reproduce asexually, microzooplankton are thought to increase in biomass faster than mesozooplankton in response to the increase in phytoplankton biomass such as occurs during the formation of a bloom (*Sherr et al., 2003*; *Sherr & Sherr, 2009*). While mesozooplankton grazing on oceanic primary production (PP) can occasionally be high (*Calbet, 2001*), microzooplankton generally tend to exert a higher grazing pressure on PP than mesozooplankton (*Calbet, 2008*; *Calbet et al., 2009*; *Campbell et al., 2009*).

Feeding by planktonic herbivores is controlled by multiple factors including feeding mode, motility, and grazer and prey sizes (*Hansen, Tande & Berggreen, 1990*; *Kiørboe, 2011*; *Wirtz, 2012*). Herbivorous protists' diverse feeding strategies (e.g., pallium feeding of dinoflagellates, *Menden-Deuer et al., 2005*) allow them to access a range of prey sizes spanning from bacteria to chain-forming phytoplankton (*Sherr & Sherr, 2002*; *Sherr, Sherr & Ross, 2013*). Nonetheless, grazing rates by herbivorous protists tend to be lower on large cells than on smaller components of the phytoplankton community (e.g., *Burkill et al., 1987*; *Verity et al., 1996*). Copepods have been shown to selectively ingest cells based on size, with lower ingestion rates for both the smallest and largest cells (e.g., *Gonçalves et al., 2014*). Feeding on large cells is, however, more efficient, as more biomass is ingested per cell, which may favor copepod feeding on large cells (*Frost, 1972*). Although large grazers such as copepods are usually considered to feed on larger particles than microzooplankton, micro- and mesozooplankton also select among similarly sized food-particles, distinguishing prey characteristics

such as nutrient composition, phylogeny, and chemical traits (*Löder et al., 2011*; *Meunier et al., 2016*).

There is increasing evidence that many copepods, including smaller species such as *Oithona* spp. and *Paracalanus spp.*, as well as naupliar stages of larger copepods such as *Calanus* spp., feed primarily on heterotrophic protists rather than on phytoplankton (*Turner, 2004*; *Calbet & Saiz, 2005*; *Campbell et al., 2009*; *Saiz & Calbet, 2011*; *Stoecker & Pierson, 2019*). Copepod feeding on these microzooplankton may suppress protistan grazing impact through top-down control on protistan abundance (*Nejstgaard, Naustvoll & Sazhin, 2001*). Such a trophic cascade can reduce all grazer-driven phytoplankton mortality, which results in higher net phytoplankton growth rates and can lead to phytoplankton biomass accumulation (*Leising et al., 2005a*; *Olson et al., 2006*; *Lavrentyev et al., 2015*). Trophic cascades can shift the fate of PP from trophic transfer to export, through the sinking of unconsumed phytoplankton due to nutrient depletion (*Schmoker, Hernández-León & Calbet, 2013*), export via fecal pellets or via vertical migration of zooplankton (*Steinberg et al., 2000*; *Gleiber, Steinberg & Ducklow, 2012*; *Siegel et al., 2016*), or physical processes such as subduction of surface waters with high phytoplankton concentrations (*Omand et al., 2015*).

Despite the recognized importance of grazing as a factor controlling phytoplankton biomass and its fate, in the North Atlantic the realized rates of grazing, the relative contribution of micro- vs. mesozooplankton, and the effects of environmental and biological conditions on grazing are poorly constrained (*Morison & Menden-Deuer, 2015*), which limits predictive capacity of these key factors in understanding the spring bloom, and ultimately its contribution to the global carbon cycle and how it may be affected by climate related changes in ocean conditions. The North Atlantic Aerosols and Marine Ecosystems Study (NAAMES) was an interdisciplinary investigation aimed at resolving key environmental and ecological processes controlling North Atlantic plankton communities, the annual cycle of PP, as well as their linkage to marine aerosols. The western North Atlantic NAAMES study region has been less studied than its eastern counterpart, and as a result is less well characterized with respect to plankton abundance, distribution, and food web dynamics (*Behrenfeld et al., 2019*). During NAAMES four campaigns, we routinely quantified rates of phytoplankton growth and microzooplankton-induced mortality, a subset of which has been published (*Morison et al., 2019*). The main objective of the work presented here was to estimate phytoplankton losses due to mesozooplankton grazing during the NAAMES spring campaign (May 2016). At three physically and biologically distinct sampling sites of the study region, in parallel with routine microzooplankton measurements, we opportunistically quantified the grazing impact of mesozooplankton, including two taxa of common north Atlantic calanoid copepods (*Calanus hyperboreus* and *Calanus finmarchicus*), and a taxonomically diverse mesozooplankton community dominated by small copepods, including *Paracalanus* spp. Light manipulation was performed at one station to investigate the role of light in the balance between phytoplankton growth and losses. Our findings suggest that feeding by large herbivorous copepods only had a minimal impact on PP but did alter the particle size spectrum, and that the effect of light availability on phytoplankton growth controlled

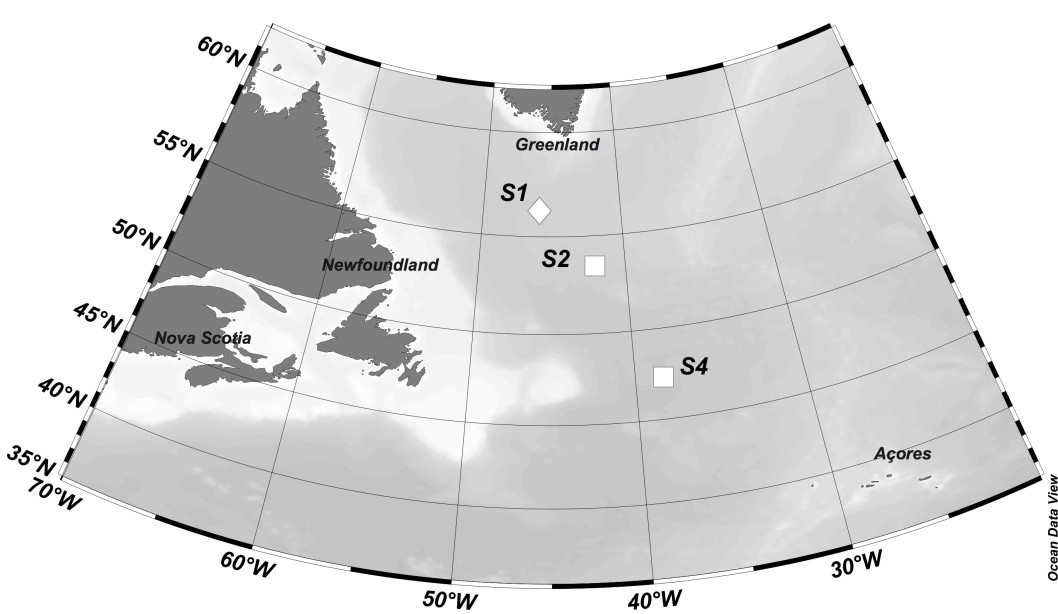

**Figure 1  Location of stations where experiments were performed.** Symbols represent eddy type (diamond = cyclonic eddy, square = anticyclonic eddy). Figure was produced using Ocean Data View software (Schlitzer, R., Ocean Data View, odv. awi.de, 2018).

phytoplankton accumulation to a greater extent than grazing by a community of small metazoan herbivores.

# MATERIAL AND METHODS

## Study area

Zooplankton grazing experiments were conducted aboard the R/V Atlantis on May 19, 21, and 27 respectively at three stations (S1, S2, and S4) located in the western North Atlantic ocean along a north to south transect (54−44°N, ∼40°W) (Fig. 1). The sequence of station numbers follows the numbering pattern during the NAAMES cruise to facilitate comparisons with other research from the field campaign. The western North Atlantic is a physically highly variable region, characterized by a complex mesoscale and sub-mesoscale eddy field (*Della Penna & Gaube, 2019*). S1 was located within a cyclonic eddy, and S2 and S4 occupied anticyclonic eddies, with S2 sampling occurring in the eddy core, and S4 sampling occurring on the eddy periphery.

## Experimental set-up

Water for the experiments was collected pre-dawn using a CTD rosette with 10 L Niskin bottles. The rosette was equipped with a SBE911plus (Seabird Electronics), a SBE43 oxygen sensor, a Wetlabs C-Star transmissometer, and a Wet Labs FLNTURTD combination fluorometer and turbidity sensor. Water collection depths were 18 m, 11 m, and 5 m at S1, S2, and S4 respectively, and were always within the mixed layer (Table 1), with mixed layer depth calculated according to the temperature algorithm as described in *Holte & Talley (2009)*. Whole seawater (WSW) was gently transferred from the Niskin bottles into 10 L

Morison et al. (2020), *PeerJ*, DOI 10.7717/peerj.9430

**Table 1  Phytoplankton growth and grazing rates in the western North Atlantic in May 2016.** Chlorophyll a (Chl a, g L$^{-1}$) concentration in source water used for the experiments (±SD), light intensity used in the incubations (% of incoming surface irradiance), and estimated rates of in situ phytoplankton instantaneous growth ($\mu$ NoN), microzooplankton (mzpkt) grazing (g), net phytoplankton growth rates in nutrient amended <200 $\mu$m seawater (k +N), non-amended <200 $\mu$m seawater (k NoN), and copepod additions (k Cop). Rates are given per day ±one standard deviation of duplicate or triplicate treatments. For copepod treatments, A, B, and C represent the different number of copepods added (A = 1, B = 3, C = 5 for *C. hyperboreu*s, A = 10, B = 20, C = 40 for *C. finmarchicus*). n/d means rate was undetermined.

| Station | Date | Chl a ($\mu$g L$^{-1}$) | Light (%) | k No Cop (d$^{-1}$) | | Copepod species | k Cop (d$^{-1}$) | | | $\mu$ NoN (d$^{-1}$) | mzpkt grazing (g, d$^{-1}$) | mzpkt % PP consumed |
|---|---|---|---|---|---|---|---|---|---|---|---|---|
| | | | | +N | No N | | A | B | C | | | |
| 1 | 5/19/16 | 2.4 (0.26) | 10 | 0.09 (0.10) | 0.02 (0.06) | *C. hyperboreus* | −0.25 (0.21) | −0.27 (0.04) | −0.40 (0.13) | 0.39 (0.09) | 0.38 (0.06) | 97 |
| 2 | 5/21/16 | 2.7 (0.11) | 10 | 0.67 (0.05) | 0.61 (0.08) | *C. finmarchicus* | 0.35 (0.26) | −0.14 (0.11) | −0.53 (0.11) | 0.61 (0.08) | 0.00 | 0 |
| 4 | 5/27/16 | 1.4 (0.09) | 40 | 0.60 (0.10) | 0.59 (0.08) | Diverse | 0.51 (0.02) | – | – | 0.59 (0.08) | n/d | n/d |
| | | | 10 | – | 0.30 (0.14) | Diverse | 0.20 (0.06) | – | – | – | – | – |
or 20 L carboys through silicone tubing fitted with 200 μm mesh over the end to exclude larger grazers.

To measure rates of microzooplankton grazing, we used the *Landry & Hassett (1982)* dilution method in a two-point modification (Fig. S1). The two-point method has been recognized as a reliable alternative to the traditional dilution series (*Chen, 2015*; *Morison & Menden-Deuer, 2017*) and has been used successfully in many studies (e.g., *Worden & Binder, 2003*; *Strom & Fredrickson, 2008*; *Landry et al., 2008*; *Landry et al., 2009*; *Landry et al., 2011*; *Lawrence & Menden-Deuer, 2012*; *Menden-Deuer, Lawrence & Franzè, 2018*). Methods are presented in detail in a companion paper (*Morison et al., 2019*). In brief, the two dilutions used were 100% and 20% <200 μm WSW. The filtered seawater (FSW) needed for the 20% WSW dilution was obtained by filtering seawater directly from the Niskin bottles through a 0.45μm membrane filter capsule (Pall). Each dilution treatment was prepared in a carboy, and gently siphoned through silicone tubing from the carboys into 1.2 L polycarbonate bottles. Duplicate bottles of each dilution were amended with a final concentration of 10 μM of both nitrate and silicate, and 1 μM of phosphate. An additional set of duplicate 100% WSW bottles was prepared without adding nutrients to serve as a nutrient control.

Mesozooplankton grazing was measured in incubations paired with the dilution assays (Fig. S1). The mesozooplankton experiments were performed using either additions of discrete numbers of handpicked copepods (S1 and S2) or a whole mesozooplankton assemblage (S4). Copepods for discrete copepod additions were collected at midnight from the upper 15 m of the water column with a vertical net tow using a 1 m diameter ring net fitted with 220 μm mesh and a non-filtering cod-end. To minimize stress on the animals, as soon as the copepod net was retrieved, the content of the cod-end was diluted into buckets containing unfiltered (S1) or filtered (S2) surface seawater. Individual undamaged copepods were selected under a dissecting microscope using wide-bore pipettes and were placed into 30 mL vials until all incubation bottles were filled. Actively swimming copepods were transferred to 1.2 L polycarbonate bottles containing <200 μm WSW. The remainder of the sample from each net tow was preserved in ethanol (10% final concentration). All fixed samples were later sorted under a stereo dissecting microscope to identify species, estimate abundances, sex, and stage distribution of copepods, accounting for those that were removed for the experiments. Animals were identified to the lowest possible taxonomic level (species for *Calanus* spp., genus for other copepods, various levels for other taxa).

Taxa used in discrete experiments were determined based on copepods observed abundance and ease of sorting. In order to measure detectable feeding rates and reduce the possibility of the copepods clearing all particles from the whole bottle, we chose the number of copepods per bottle based on expected per capita clearance rates, such that the copepods in the bottles would not be expected to clear more than 30–40% of the volume in the bottles (*Gifford, 1993*). A range of copepod concentrations was used to bracket the potential variability in feeding rates. At S1, feeding experiments were conducted using *C. hyperboreus* despite the species not being the most abundant, because based on its reported distribution range, it was expected that its prevalence would decrease at more southern
stations, preventing further investigation. Adult individuals, known to range in size from 5–7 mm (*Leinaas et al., 2016*), were added to the incubation bottles at concentrations of 1, 3, or 5 individuals per bottle. At S2, adult individuals of *C. finmarchicus* were used. Based on this species' smaller size (2.5–2.7 mm; *Leinaas et al., 2016*), 10, 20, and 40 individuals $L^{-1}$ were added. At S4, net tows revealed the dominance of small (<1 mm) metazoa that would be difficult to sort by hand, as done for S1 and S2. Hence some whole seawater was collected without the 200 $\mu$m screening mesh so as to include mesozooplankton. Incubations of this unscreened whole seawater served as the experimental treatment representing the mesozooplankton assemblage at concentrations occurring *in situ*.

Discrete copepod additions were incubated in duplicate for each copepod concentration, and unscreened WSW including the mesozooplankton assemblage used at S4 was incubated in triplicates (Fig. S1). In all experiments, no nutrients were added to the bottles containing the mesozooplankton. Incubations for all experiments lasted 24 h so as to encompass grazers' possible diel feeding cycle. Bottles were placed in 250 L on-deck plexiglass incubators kept at ambient temperature using an open-circuit flow of surface seawater. Neutral density screening was used to maintain incubations at light intensities targeted to correspond to *in situ* light intensity at the collection depth, approx. 10% of surface irradiance at S1 and S2, and 40% of surface irradiance at S4. At S4, to ensure compatibility with S1 and S2, to estimate potential high irradiance effects on mesozooplankton, and to quantify the light dependence of phytoplankton growth and mortality due to mesozooplankton grazing, additional duplicate bottles of <200 $\mu$m WSW and triplicate bottles of unscreened WSW were incubated without added nutrients at an additional light intensity of 10% surface irradiance, which corresponded to the *in situ* light at a ∼20 m depth. The percentage of light reduction achieved with the neutral density screening was measured with a QSL-2100 Scalar irradiance sensor (Biospherical Instruments Inc.) and verified using a Hobo (onset) data logger placed in the incubators that recorded light intensity at 1 min intervals. To keep all bottles in an experiment exposed to an even light intensity, bottles were suspended from a line running across the incubators, which, along with the combined effect of ship motion and flow of seawater through the incubators, provided gentle agitation.

## Rate estimates

Zooplankton grazing rates were estimated from changes in chlorophyll a (Chl a) over the incubation duration. For each experiment, Chl a concentration (P) was determined from triplicate 180 mL subsamples of initial stock ($P_0$) and of each replicate incubation bottle ($P_t$) filtered over 0.7 $\mu$m glass microfiber filters (Whatman GF/F). Chl a determination followed *Graff & Rynearson (2011)*, except that extraction took place at room temperature and in the dark for 12–15 h in 96% ethanol (*Jespersen & Christoffersen, 1987*).

Changes in light conditions between *in situ* and incubations can induce phenotypic responses of phytoplankton cells known as photoacclimation (*Gutiérrez-Rodríguez et al., 2010* references therein). The photoacclimation process can introduce artifacts in the estimation of phytoplankton growth rates due to changes in phytoplankton cells' pigment content and therefore in the Chl a to carbon ratio in response to a change of light

(*Ross et al., 2011*). To avoid such artifacts, a photoacclimation index (Phi) was estimated following *Gutiérrez-Rodríguez et al. (2010)*, *Gutierrez-Rodriguez et al., (2011)* and *Morison et al. (2019)*, using flow cytometry (FC) data obtained from 200 µL aliquots of <40 µm screened WSW analyzed live on a Guava® easyCyte Flow Cytometer. FC data were collected as previously described in *Morison et al. (2019)*. Samples were run at 0.24 µl s$^{-1}$ for 3 min. Three phytoplankton groups (*Synechococcus* spp., pico- and nanoeukaryotes) were distinguished based on their forward scatter and red (695/50) emission parameters with 488 nm excitation, and orange (620/52) emission parameters with 532 nm excitation. For each sample, we calculated a photoacclimation index (Phi) from initial and final FC measurements of red fluorescence to forward scatter ratio (FLR:FSC), the latter being used as a proxy for Chl a:carbon (*Graff & Behrenfeld, 2018*). Phi values <1 indicate a decrease in the FRL:FSC ratio (i.e., less Chl a per unit of biomass), values >1 indicate an increase in the ratio (more Chl a per unit of biomass), and values = 1 indicate no change.

Rates of phytoplankton growth and microzooplankton grazing were estimated as previously described in *Morison et al. (2019)*. Net phytoplankton growth rate ($k$, d$^{-1}$) in each experimental bottle was estimated using the equation $k = 1/t \times \ln([P_t/Phi]/P_0)$, where t is the duration of the incubation in days, and with final Chl a concentration ($P_t$) corrected for photoacclimation-derived changes in cell pigment content using Phi. For S4, missing data prevented estimation of Phi, thus rates for that station were not adjusted for photoacclimation. Microzooplankton grazing rates ($g$, d$^{-1}$) were estimated using $k$ values from nutrient amended replicates using the equation $g = (k_{dil} - k_1)/(1 - D)$ in which the subscripts *dil* and 1 correspond to the nutrient-amended diluted and undiluted treatments respectively, and D represents the achieved fraction of WSW in the diluted treatment, as determined from measured initial Chl a concentration. A model I linear regression of $k$ as a function of dilution was used to determine whether grazing was significantly different from zero ($\alpha \leq 0.05$), and to yield the standard error of the grazing rate estimate. Phytoplankton *in situ* specific growth rates ($\mu$, d$^{-1}$) were determined as the sum of the grazing rate ($g$, d$^{-1}$) and the mean net growth rate of the non-amended <200 µm WSW bottles. Negative values of grazing rates, which result when the phytoplankton apparent growth rate ($k$) is lower in the diluted than in the undiluted treatments, indicate a violation of a central assumption of the dilution method (*Landry & Hassett, 1982*). Thus in case of statistically significant negative grazing rates, losses were undetermined, and in the absence of a grazing loss estimate, $\mu$ was equated to the net growth rate ($k$) in the undiluted non-amended bottles.

For S1 and S2, copepod clearance rates for *C. hyperboreus* and *C. finmarchicus* based on bulk Chl a measurements were calculated as described previously (*Leising et al., 2005b*). Briefly, the net phytoplankton growth rates ($k$) in the copepod additions and the microzooplankton-only control were regressed against the concentration of copepods in each treatment, with the 100% <200 µm seawater treatment from the microzooplankton dilution experiments used as "zero" copepod treatment. The slope of the regression line is equivalent to the per capita daily grazing rate of copepods on Chl a. Negative slopes of this regression indicate removal of Chl a through copepod feeding. Specific grazing rates were converted to clearance rates using the relationship of $F = Vg/N$, where $F$ is the clearance rate (mL individual$^{-1}$ d$^{-1}$), $V$ is the volume of the experimental container (mL), $g$ is the

grazing rate ($d^{-1}$), and N is the number of copepods (*Frost, 1972*). Copepod viability was verified for all incubation bottles at the end of the experiments. Corrections for copepod mortality were not necessary, because all individuals except one *C. hyperboreus* in a bottle with 5 copepods were found to be actively swimming at the end of the incubation period.

For S4, total phytoplankton mortality due to the combined effects of micro- and mesozooplankton grazing was calculated by subtracting the net phytoplankton growth rate ($k$) in the unscreened WSW bottles from $\mu$. Mesozooplankton grazing rates were then calculated as the difference between total and microzooplankton-induced mortality rates.

In order to estimate grazing effects on particle size abundance spectra, particle abundances between 3 and 60 $\mu$m equivalent spherical diameter (ESD) were determined from two mL samples collected from initial stock and from all bottles at the end of the incubation period and analyzed with a Multisizer$^{TM}$ 3 Coulter Counter$^{\circledR}$ (Beckman Coulter, USA). A FSW sample was also analyzed and served as a correction blank. Blank controls contained a negligible amount of particles, with a coefficient of variation of 1%. Significant difference in particle size abundance spectra for the copepod treatments relative to the microzooplankton-only control was assessed with a 2-sample Kolomogorov-Smirnov test at a significance level of $p < 0.05$.

## RESULTS

Each station had distinctive physical characteristics reflective of its latitudinal location and the mesoscale feature it occupied (Fig. 1). Mixed layer depth was 56 m, 41m, and 22 m at S1, S2, and S4 respectively. Mixed-layer average water temperature ranged from 3.9 °C at S1 to 15.5 °C at S4 (Fig. 2). Mixed-layer average salinity ranged from 34.6 at S2 to 36.1 at S4 (Fig. 2). Chl a concentration in the source water, i.e., *in situ* concentration at sampling depth for the incubation experiments, ranged from 1.4 ($\pm$0.09) $\mu$g L$^{-1}$ at S4 to 2.7 ($\pm$0.11) $\mu$g L$^{-1}$ at S2 (Table 1). Collection depth for source water and copepods was within the surface mixed layer, well shallower than the mixed layer depth (Fig. 2). Copepod community composition differed between stations (Fig. 3). *C. finmarchicus* was present at all stations, and was numerically dominant at S1 and S2 (Table S1). *C. hyperboreus* was only present at S1. The most diverse assemblage was collected at S4, dominated by *C. finmarchicus*, *Paracalanus* spp., and *Pleuromamma* spp., but also including *Pseudocalanus* spp., *Metridia lucens*, and several other taxa.

### Effect of micro- and mesozooplankton grazing on phytoplankton growth

Microzooplankton grazing was limited to the most northern station (S1), where microzooplankton grazing rate was $0.38 \pm 0.06$ d$^{-1}$ and phytoplankton specific growth rate ($\mu$) was $0.39 \pm 0.09$ d$^{-1}$. Thus micro-grazers at S1 consumed almost all (97%) PP (Table 1). No significant microzooplankton grazing was detected at S2 (Table S2), where $\mu$ was $0.61 \pm 0.08$ d$^{-1}$. At S4, microzooplankton grazing measurement was conducted at the *in situ* light level only. The net phytoplankton growth rate ($k$) in the 20% WSW dilution was lower than in any of the WSW treatments (Table S2), representing a violation of one of the method's central assumptions, i.e., that $\mu$ is constant across dilutions (*Landry*
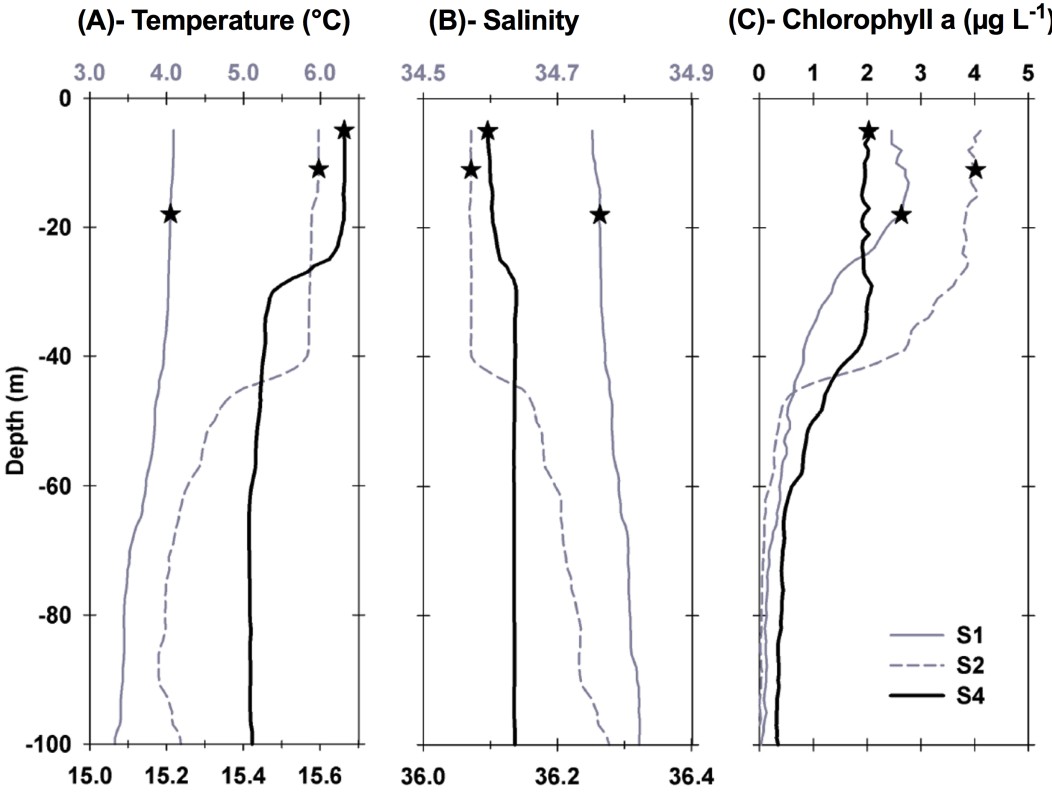

**Figure 2  Vertical profiles of physical and biological properties at each sampled station.** Profiles shown include (A) temperature, (B) salinity , and (C) chlorophyll a. Depths, at which water for the experiments was collected, are shown as stars. For best profile definition, temperature and salinity are plotted on the top $x$-axis for S1 and S2 and on the bottom $x$-axis for S4. Line style for each variable is the same, with S1 represented by grey solid lines, S2 represented by grey dashed lines, and S4 by thick black lines.

*& Hassett, 1982*). Thus at S4, the microzooplankton grazing rate could not be determined, and the net growth rate in the <200 μm WSW without added nutrients ($0.59 \pm 0.08$ d$^{-1}$) was considered to represent in situ μ.

At both S1 and S2, mesozooplankton feeding on phytoplankton was indicated by the decreasing net phytoplankton growth rate ($k$) with increasing numbers of copepods (Fig. 4). At S1, $k$ was significantly inversely related to grazers' abundance (Model I regression, $df = 6$, $p = 0.027$). Net phytoplankton growth was highest in the incubation where only microzooplankton grazers were present ($0.01 \pm 0.07$ d$^{-1}$), and decreased with increasing number of copepods added, from $-0.25 (\pm 0.21)$ d$^{-1}$ when one individual of *C. hyperboreus* was added to $-0.40 (\pm 0.13)$ d$^{-1}$ when five individuals were added (Fig. 4A). Similarly, at S2, $k$ was highest when only the microzooplankton fraction was considered ($0.61 \pm 0.08$ d$^{-1}$) and decreased significantly (Model I regression, $df = 6$, $p = 0.0002$) with increasing copepod concentrations, ranging from $0.35 \pm 0.26$ d$^{-1}$ for 10 individuals of *C. finmarchicus* to $-0.53 \pm 0.11$ d$^{-1}$ when 40 individuals were added (Fig. 4B).

Per capita grazing rates were calculated for stations S1 and S2, where experiments were conducted with individual copepods sorted and placed into experimental bottles. For *C.*

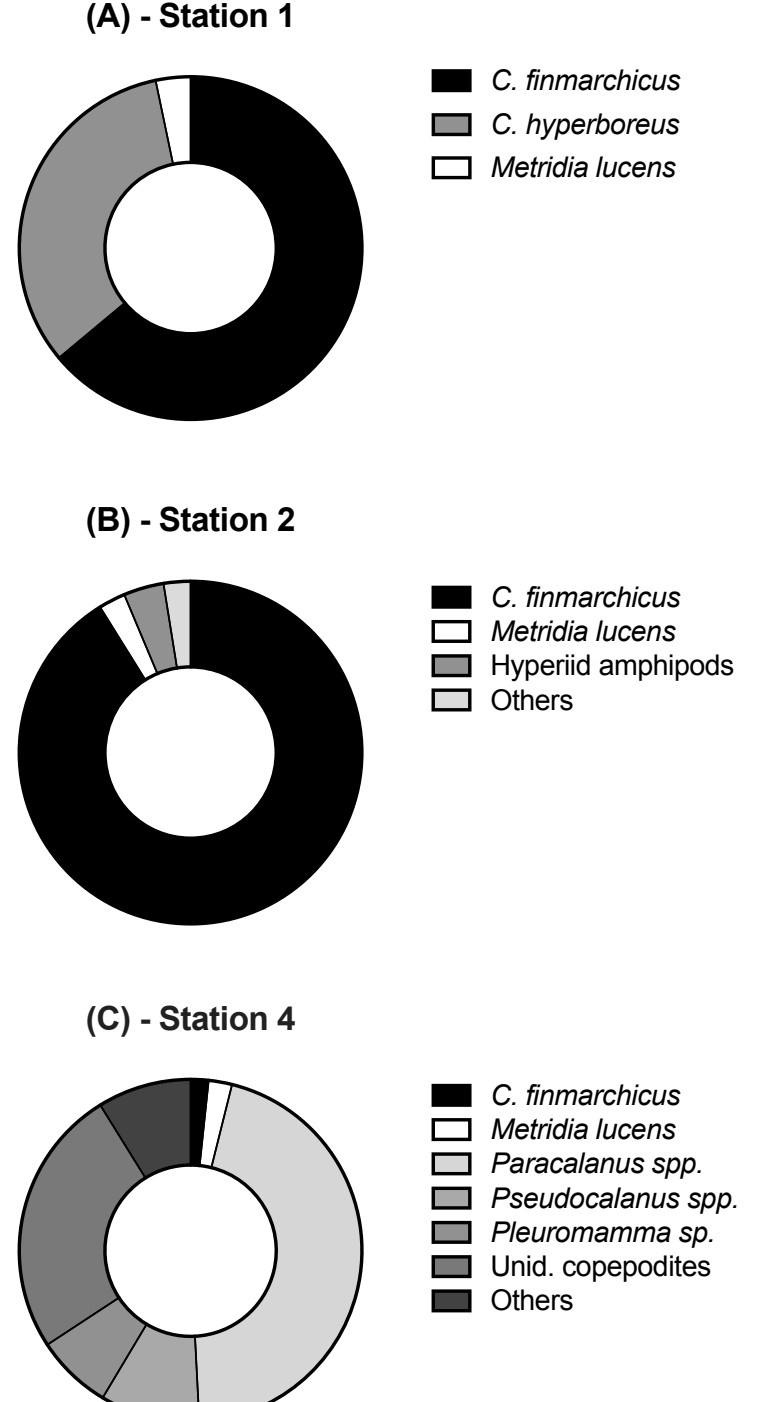

**(A) - Station 1**

- ■ *C. finmarchicus*
- ▨ *C. hyperboreus*
- □ *Metridia lucens*

**(B) - Station 2**

- ■ *C. finmarchicus*
- □ *Metridia lucens*
- ▨ Hyperiid amphipods
- □ Others

**(C) - Station 4**

- ■ *C. finmarchicus*
- □ *Metridia lucens*
- ▨ *Paracalanus spp.*
- ▨ *Pseudocalanus spp.*
- ▨ *Pleuromamma sp.*
- ▨ Unid. copepodites
- ▨ Others

**Figure 3** **Relative abundance of copepod and mesozooplankton species found at the three sampled stations.**

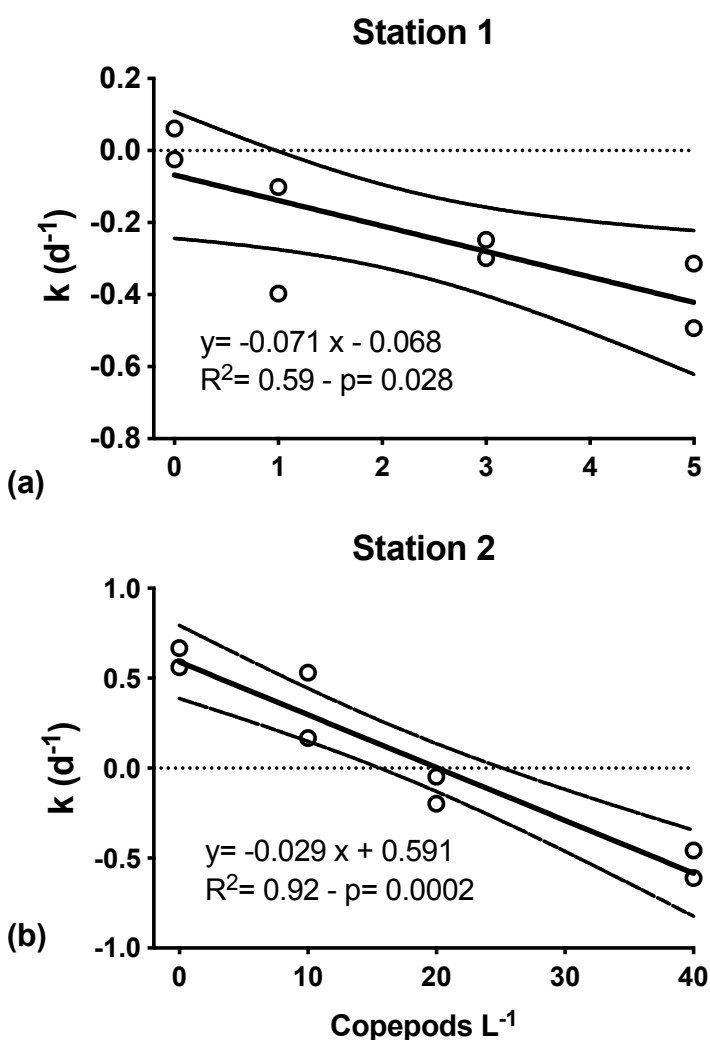

**Figure 4** **Net phytoplankton growth rates (k, d⁻¹) as a function of discrete additions of copepods.** (A) *C. hyperboreus*, station 1. (B) *C. finmarchicus*, station 2. Zero copepod treatments correspond to <200 μm undiluted seawater samples from microzooplankton dilution experiments. Significant negative slopes indicate copepod removal of phytoplankton biomass, resulting in reduced net phytoplankton growth. Dashed lines represent slope 95% confidence interval.

*hyperboreus* mean (± standard error) grazing was 0.071 (±0.024) d⁻¹ at S1 and for *C. finmarchicus* it was approximately half at 0.029 (±0.004) d⁻¹ at S2 (Fig. 4). Conversion of specific grazing rates resulted in clearance rates of 64.2 (±2.2) mL individual⁻¹ d⁻¹ for *C. hyperboreus* and 30.4 (±4.2) mL individual⁻¹ d⁻¹ for *C. finmarchicus*.

At S4, the relative effect of light vs. mesozooplankton grazing on net phytoplankton growth (k) was quantified by incubating both the microzooplankton (<200 μm WSW) and the whole community of mesozooplankton (unscreened WSW) under a 10% light intensity representative of a depth of ∼20 m, in parallel with the experiment conducted at surface irradiance. Realized mean light intensity in the incubations in the two light treatments was 53% (±5% SD of the mean of 1 min interval measurements) and 12% (±1% SD) of

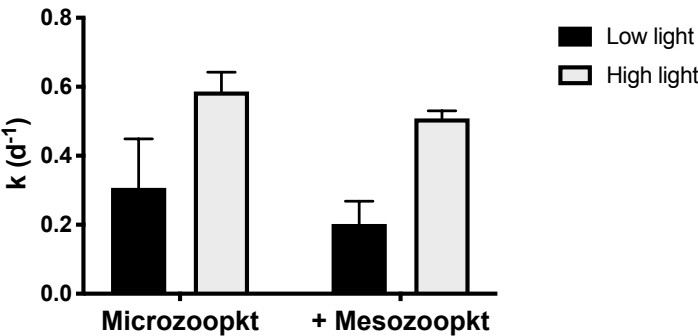

**Figure 5** **Net phytoplankton growth rates (k, d⁻¹) at Station 4 in relation to light treatment and grazer's type.** Rates are given as mean ±SD of duplicate (microzooplankton) and triplicate (mesozooplankton) measurements. Low and high light refer to realized light intensities of 10% and 50% surface irradiance respectively.

surface irradiance. Both the consumer group present and the light intensity used in the incubation had an effect on $k$, but the largest effect was observed in response to variations in light intensity (Fig. 5). Irrespective of grazer treatment, $k$ was significantly lower (two-way ANOVA, $p = 0.0009$) in incubations at the 12% irradiance than at the higher irradiance (Table S3), by 49% in the microzooplankton-only treatment and 61% in the copepod addition (Table 1). Irrespective of light treatment, addition of mesozooplankton reduced $k$ by approx. $0.1$ d⁻¹ compared to treatments with microzooplankton only (Table 1) but the effect was not significant (two-way ANOVA, $p = 0.105$; Table S3). Incubations of whole seawater left unscreened, so as to include mesozooplankton at their *in situ* abundances, conducted at a light intensity corresponding to the depth of sample collection, resulted in a 14% reduction in $k$ relative to the <200 μm control, corresponding to a removal of 14% of PP (Table 1). There was no significant interaction between light intensity and grazer size fraction ($p = 0.793$).

## Particle removal by micro- and mesozooplankton

An effect of copepod grazing on the size abundance spectrum of particles in the 3–60 μm size range was observed, indicating a shift in size spectra mediated by different grazer communities. Although few particles >10 μm were generally observed, a larger proportion of >10–15 μm particles were removed by mesozooplankton grazers at all stations over the course of the 24 h incubations, in comparison with microzooplankton-only incubations (Fig. 6). The average coefficient of variation among copepod/mesozooplankton replicates was 11–13% for particles 3–20 μ, with minimum values of 0–3%. Measurably higher particle abundances in the >10 μm, and particularly in the 10-20 μm size range were observed in microzooplankton-only incubations compared to incubations that contained mesozooplankton grazers. The effect of copepod grazing on particle size distribution was greatest at S2 (Fig. 6B), where at concentrations of 20 and 40 *C. finmarchicus* per bottle, *ca.* 5-fold lower particle abundances in the 10 to 20 μm size range remained after incubation. In contrast, at that same station there was no measurable change in the relative contribution of each size category to total particle abundance in the microzooplankton-only incubations,
and size spectra of the initial and final samples were indistinguishable (Fig. 6B). This observation is consistent with the lack of detectable microzooplankton grazing at that station.

At S1, the difference between treatments in particle size distribution (PSD) was not significant, either for the entire spectrum (Kolmogorov/Smirnov test, $p = 0.62$) or for >5 µm ($p = 0.56$) and >10 µm ($p > 0.25$) particles. However, the analysis was done for the treatment with only one copepod added per bottle, which was likely insufficient to manifest a significant change in particle concentration. At S2, PSD in the copepod treatment was significantly different from the microzooplankton-only treatment ($p = 0.04$) for particles >5 µm when 40 copepods per bottle where present, and for the $\geq$ 5 µm to 20 µm size range when only 20 copepods were added. At S4, although variable among size bins, the net rate of change in the abundance of particles in the microzooplankton treatment was similar to the Chl a based net growth rate from the dilution experiment, with an average of 0.6 d$^{-1}$ for particles 3–30 µm, a size fraction representing 99% of the total counts. Although an increase in particle abundance was observed for most size bins irrespective of the type of grazer, the number of >5 µm particles in the mesozooplankton treatment were reduced relative to the microzooplankton control (Fig. 6C) and the difference in PSD between the two treatments was significant ($p = 0.04$).

## DISCUSSION

In many systems, the degree of grazer-induced phytoplankton mortality and the type of grazer largely drive the fate of primary production (*Steinberg & Landry, 2017*). Whether organic matter is shuttled to higher trophic levels, recycled within the photic zone, or exported to depth via fecal pellets or other mechanisms, depends on the relative grazing impact of large metazoa and herbivorous protists. Zooplankton grazing is also considered to play a role in the development of phytoplankton blooms (*Sherr & Sherr, 2009*), including the formation of the North Atlantic spring bloom (*Behrenfeld & Boss, 2014*). In this study, we observed a spatial switch in the relative grazing impact of micro- vs. mesozooplankton. At the most northern S1, microzooplankton were responsible for a high removal of phytoplankton production, whereas larger, mm-sized copepods had a low grazing impact, suggesting a relatively greater rate of biomass recycling in the surface compared to export facilitated by larger specimen that produce significant fecal pellets and/or vertically migrate (*Ducklow, Steinberg & Buesseler, 2001*; *Schnetzer & Steinberg, 2002*). At the more southern S2 and S4, grazer-induced losses of primary production were low and essentially due to mesozooplankton. At S2, grazing by microzooplankton was undetectable, and although grazing was undetermined at S4, the high net growth rates of a doubling per day obtained at that station both from Chl a and particle counts suggest that microzooplankton grazing was minimal. Remarkably, the expected decrease in net phytoplankton growth rates under reduced light at S4 was approximately three times larger than the effect of grazing. The observations presented here may well be restricted to the specific conditions encountered during the study. However, the biological and physical characteristics of the three stations span a considerable dynamic range and the results provide concrete rate estimates and

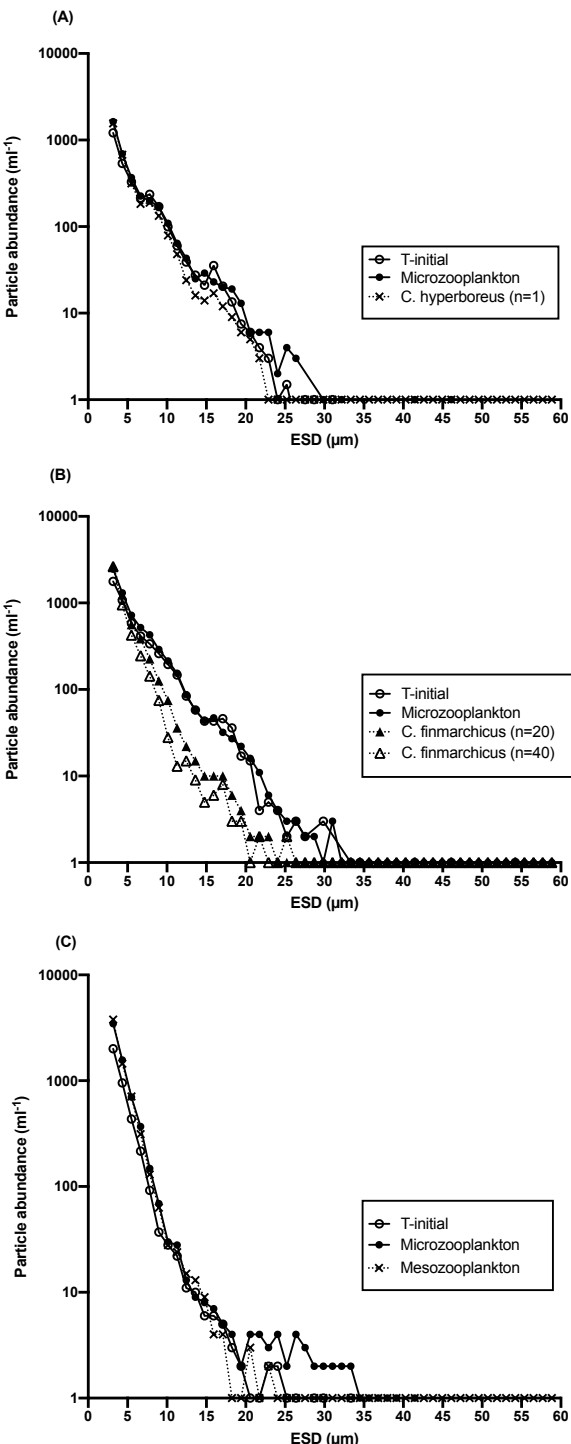

**Figure 6** **Particle abundance (mL$^{-1}$) vs. size as equivalent spherical diameter (ESD), μ m) at the beginning and at the end of 24 h incubations with and without mesozooplanktkon.** (A) Station 1, one individual of *C. hyperboreus*, (B) Station 2, 20 and 40 individuals of *C. finmarchicus* and (C) Station 4 whole mesozooplankton community (>200 μ m) at *in situ* light intensity (~40% of surface irradiance). For clarity, error bars for replicated samples have been omitted. The variation among replicates depended on size class, with higher variation in the higher size bins, due to low (<5 particles mL$^{-1}$) abundance.

testable hypotheses linking physical features, grazing pressure, light availability, and rates of transfer of matter and energy in the western North Atlantic, where planktonic food web dynamics are poorly characterized.

Clearance rates of the large copepods investigated here can vary widely depending on bloom stage, prey availability, and prey type (e.g., *Meyer-Harms et al., 1999*; *Levinsen et al., 2000*). The clearance rates presented here fall within the lower end of the published range (e.g., *Huntley & Tande, 1987*; *Hansen, Tande & Berggreen, 1990*; *Nejstgaard et al., 2001*; *Mayor et al., 2006*), although similar average rates of 25.2 ml cop$^{-1}$ d$^{-1}$ have previously been recorded in the same season and location -May in the North Atlantic- for *C. finmarchicus* (*Gifford et al., 1995*). Calculations of the direct impact of *C. hyperboreus* and *C. finmarchicus* on PP are problematic with our net tows, which were not designed to quantitatively estimate abundance of these taxa. Nonetheless, considering the clearance rates estimated here, it would require more than 8,000 m$^{-3}$ *C. hyperboreus* and 17,000 m$^{-3}$ *C. finmarchicus* to clear 100% of the PP. Even at triple our measured clearance rates, removal of 100% PP would require much larger copepod concentrations than are generally reported from North Atlantic offshore ecosystems in late spring/early summer (e.g *Scott et al., 2000*; *Madsen, Nielsen & Hansen, 2001*; *Heath et al., 2004*; *Castellani et al., 2008*)—although not entirely impossible. Copepods in general tend to spatially aggregate and form patches (*Genin et al., 2005*; *Hirche et al., 2006*), and peak abundances of mono-specific aggregations can be up to two orders of magnitude higher than in the water column (*Scott et al., 2000*; *Madsen, Nielsen & Hansen, 2001*; *Hirche et al., 2006*; *Castellani et al., 2008*). Although it would be locally restricted, within these patches a higher grazing impact on phytoplankton biomass would be expected, and thus a greater local impact on PP by copepods. It is difficult to assess the quantitative impact of such grazer patches as sizes and frequencies are poorly constrained and variable over time (*Basedow et al., 2013*).

Previous studies on the grazing impact of *C. finmarchicus* and *C. hyperboreus* under bloom conditions were often focused on polar waters and estimates have varied considerably among sites. For example, the grazing impact of a *Calanus* dominated community was estimated to range between 15–93% of PP in the upper 50 m of Disko Bay during the main bloom period (*Madsen, Nielsen & Hansen, 2001*). In arctic waters, where production is highly restricted seasonally, *C. hyperboreus*, and *C. glacialis* grazed 65–90% of the phytoplankton carbon production (*Eilertsen, Tande & Taasen, 1989*). Occasionally, copepods such as *C. finmarchicus* can suppress bloom development (*Williams & Lindley, 1980*; *Tiselius, 1988*), but grazing of *C. finmarchicus* and *C. hyperboreus* may not affect PP if their biomass is low (*Hirche et al., 1991*). Studies focusing on grazing by *C. finmarchicus* have shown the species to have a low grazing impact of 1–10% of PP (*Cowles & Fessenden, 1995*; *Gifford et al., 1995*).

Although these conclusions come from studies that have focused on particular copepod species, similar conclusions have been drawn from studies of whole mesozooplankton communities. In the North Atlantic ocean, assessments of mesozooplankton grazing during spring bloom events have largely focused on the northeast region (*Morales et al., 1991*; *Dam, Miller & Jonasdottir, 1993*; *Lenz, Morales & Gunkel, 1993*; *Barquero et al., 1998*; *Halvorsen et al., 2001*), and to the best of our knowledge no data are available for the

northwest region, where the present study was conducted. The majority of these studies concluded that phytoplankton biomass in the eastern North Atlantic is underexploited by mesozooplankton communities, with mesozooplankton grazing removing between 3% (*Dam, Miller & Jonasdottir, 1993*) and 14% PP (*Halvorsen et al., 2001*). Most studies relied on gut fluorescence analysis and their results are not directly comparable to the methodology applied in our study. In a methodologically similar study to ours using parallel incubations of dilution experiments with and without added copepods, *Campbell et al. (2009)* measured approximately 13% removal of PP by copepods in the western Arctic Ocean. Thus, irrespective of the methods used, our results and these prior assessments agree, and all lead to the conclusion that *Calanus* species have a small impact on PP and are not usually able to suppress bloom development in the North Atlantic.

It has been argued that copepod grazing impact has commonly been underestimated due to the use of nets >200 μm that leave out smaller species and stages of copepods (*Turner, 2004*). In our experiments at S1 and S2, copepods were collected using a 220 μm net, thus the grazing impacts presented here are based on the species investigated, assuming their prevalence in the >220 μm mesozooplankton community, and may represent underestimates of the impact on phytoplankton by metazoan grazers as a whole. Some of the smaller <200 μm metazoan specimen are part of the microzooplankton community (*Calbet, 2008*), however the dilution method used to quantify microzooplankton grazing does not allow to distinguish between the small metazoan grazing impact and that of protistan grazers.

The low impact of grazing by copepods on phytoplankton globally has been interpreted as an indication that copepods likely rely on other food sources to supplement their diet (*Calbet, 2001*; *Saiz & Calbet, 2011*), and indeed microzooplankton can be an important component in the diet of mesozooplankton (*Stoecker & Pierson, 2019*). High grazing rates and selective feeding by mesozooplankton on microzooplankton in incubations would release microzooplankton grazing pressure on phytoplankton, and thus could lead to an increase in net phytoplankton growth compared to the microzooplankton-only treatments and to an underestimation of copepod grazing rates (*Nejstgaard, Gismervik & Solberg, 1997*; *Nejstgaard, Naustvoll & Sazhin, 2001*). Here, however, we did not observe the induction of such trophic cascades where higher net phytoplankton growth rates are observed in the presence of copepod grazers, which would have suggested preferential feeding on herbivorous protists. Indeed our calculation of clearance rates indicated direct uptake of phytoplankton for both S1 and S2, which scaled with the concentration of copepods. It is possible that copepods feeding on both phytoplankton and microzooplankton would offset the individual effect on net phytoplankton growth rates of each feeding mode, but at S1, the only station where microzooplankton grazing was significant, such offset would fail to explain the large differences in net phytoplankton growth between the copepod treatments and the control. It is likely that copepod feeding was omnivorous, and microzooplankton and phytoplankton were grazed by mesozooplankton at proportions similar to their relative abundances (*Barquero et al., 1998*; *Halvorsen et al., 2001*).

Despite their low feeding impact on the phytoplankton community as a whole, copepods had a remarkable impact on the particle size spectrum. Although few particles were observed

in the >10 μm size range, these particles were removed only in the presence of copepods. This observation supports the often-observed feeding preference of Calanoid copepods for larger particles (e.g., *Frost, 1972*; *Gifford et al., 1995*; *Levinsen et al., 2000*). Particles <5 μm tend to be too small for these copepods, although very large particles such as diatoms with spines may be too large (*Campbell et al., 2009*). Copepods, however, demonstrate extensive flexibility in their diet (*Kleppel, 1993*) and may feed size-selectively when food is abundant, and non-selectively when food becomes scarce (*Cowles, 1979*). It is possible that if copepods in our incubations fed preferentially on large prey types, these may have been depleted before the end of the experiment, potentially biasing feeding rate estimates. However, due to the low in situ abundance of large particles, feeding on such particles would also be limited in the field. As observed here, size-selective copepod grazing can induce shifts in the size distribution of the plankton community, which may have important implications regarding the potential for particles gravitational sinking and the associated vertical flux of carbon (*Stemmann & Boss, 2012*).

In most instances when comparisons between copepod and microzooplankton grazing are possible, the grazing impact by the unicellular herbivores typically exceeds that of the mesozooplankton component at least 2–3 fold (e.g., *Morales et al., 1991*; *Burkill et al., 1993*; *Dam, Miller & Jonasdottir, 1993*; *Weeks et al., 1993*; *Gifford et al., 1995*). In our study, however, the relative grazing impact of both types of grazers varied spatially. At S1, microzooplankton grazing removed 97% of PP, exceeding the ∼66% estimated global average of the proportion of PP removed by herbivorous protists (*Calbet & Landry, 2004*). Perhaps due to their diverse feeding modes, microzooplankton can indeed have a considerable impact on PP. This potential has been documented in the mixed layer of the northeast Atlantic under bloom (*Verity et al., 1993*) and non-bloom conditions (*Burkill et al., 1993*), during developing blooms in the polar and subpolar Northeast Atlantic (*Morison & Menden-Deuer, 2015*; *Menden-Deuer, Lawrence & Franzè, 2018*), and in post-bloom conditions (*Gifford et al., 1995*). Estimates of the aforementioned studies on daily PP removal by microzooplankton ranged from 15–242%, with the highest average impact (81%) being attributed to after-bloom conditions (*Gifford et al., 1995*).

In contrast to the high microzooplankton grazing impact at S1, no microzooplankton grazing was observed at S2 and remained undetermined at S4. Absence of grazing is not uncommon, and has been recorded in all ocean ecosystems, from estuaries to the coastal and open ocean, and in all ocean basins at all latitudes (*Schmoker, Hernández-León & Calbet, 2013* and references therein). Lack of grazing could suggest that grazers' biomass had not sufficiently accumulated to have a detectable effect on PP as hypothesized by *Sherr & Sherr (2009)*. Such biomass buildup can be delayed if grazers have been exposed to prolonged periods of starvation (*Anderson & Menden-Deuer, 2017*), which can occur during winter when phytoplankton biomass is low, or when mixing of the ocean surface dilutes plankton populations (*Morison et al., 2019*). Lack of grazing could also be the result of a mismatch between prey species and grazer feeding types, which can be brought about by frequent physical disturbances occurring in the North Atlantic (*Morison & Menden-Deuer, 2015*; *Morison et al., 2019*). Alternating observations of high grazing impact and of no-grazing have been documented in numerous studies, suggesting an "all or nothing" grazing impact

may be characteristic of microzooplankton grazing dynamics (*Menden-Deuer, Lawrence & Franzè, 2018* and references within). The negative microzooplankton grazing rates obtained at S4 prevented any estimation of microzooplankton grazing impact. It has been suggested that the presence of chloroplasts in mixotrophs can artificially increase the Chl a-based apparent growth rate in the undiluted treatment, resulting in a positive slope (*Landry, Constantinou & Kirshtein, 1995*; *Calbet et al., 2012*). Unfortunately the impact -if any- that this process may have had on grazing rate estimation at S4 is difficult to evaluate, as assessing the proportion of mixotrophs in a plankton assemblage that engage in phagotrophy remains challenging (*Beisner, Grossari & Gasol, 2019*).

Although our experimental design controlled for light effects on mesozooplankton-grazing rates, no such effect was observed in our experiments at S4. Many copepods exhibit diel periodicity in their feeding (*Durbin, Durbin & Wlodarczyk, 1990*; *Durbin et al., 1995*), that would negate some effects of light, however, under some conditions copepods may feed during daytime (*Atkinson et al., 1992*). Nonetheless, our observations contrast with previous studies that have shown significantly higher copepod feeding rates at lower irradiances, indicating that light cues may influence both the timing of grazing and the gut fullness in certain copepod species (*Stearns, 1986*; *Cieri & Stearns, 1999*). In contrast, as expected, there was a significant effect of light on phytoplankton growth rates, both in the presence and absence of copepods. Due to differences in phytoplankton growth in response to different light intensity, the mesozooplankton grazing impact on PP estimated as the fraction of total growth removed was almost three times greater at lower light. Net phytoplankton growth rates at S4 were not adjusted for photoacclimation. Thus at the lower light intensity, net phytoplankton growth rates may have been over-estimated, if a decrease in light resulted in phytoplankton increasing their cellular pigment content. This lack of adjustment, however, does not change our conclusions that light did not influence mesozooplankton-grazing rates. Indeed any photoacclimation process would have affected net phytoplankton growth rates in all treatments similarly, and thus differences between the copepod treatments and the controls with or without adjustment would remain unchanged. On the other hand, if the net phytoplankton growth rates were overestimated at the low light intensity, then the difference in net growth rates between the high and the low light treatments may have been underestimated, magnifying the greater effect of light than of grazing on phytoplankton accumulation. The differential effect of light on growth and grazing rates observed here suggests that in the ocean, the exponential decrease of light with depth acts to decouple growth and grazing, resulting in a vertically increasing grazing impact on PP.

## CONCLUSIONS

Concurrent investigation of mesozooplankton and microzooplankton grazing impact on phytoplankton primary production in the western North Atlantic suggested a generally limited potential control of phytoplankton biomass by herbivorous copepods, sometimes but not always exceeded by microzooplankton grazing. Although the observations reported here were acquired opportunistically and are limited in sampling scope, they represent a

diverse range of environmental conditions and species composition from a poorly sampled region. The generally low mesozooplankton grazing impact across stations despite the spatial diversity of species and conditions, as well as the good agreement with similar studies in other regions, provide important constraints to quantifying grazer-induced phytoplankton mortality rates for this region and the NAAMES spring campaign. There was no indication of preferential feeding on microzooplankton by larger copepods. Remarkably, when compared, light effects on net phytoplankton growth rates exceeded mesozooplankton grazing effects by at least 3-fold. Together these coupled biological and environmental effects provide insights into the transfer and recycling of recently fixed carbon. Specifically, they suggest limited vertical export of phytoplankton production as mediated by large zooplankton. Such concurrent environmental and biological process data on controls of primary production provide important parameters for modeling frameworks assessing global elemental fluxes and carbon export.

## ACKNOWLEDGEMENTS

The authors appreciate the support from the Captain and Crew of the R/V Atlantis during cruise AT34 in 2016 and leadership from Chief Scientist Mike Behrenfeld (Oregon State University). We thank Peter Gaube and Alice Dellapenna (University of Washington Applied Physics Lab) for sharing their assessment of the eddy field, Pierre Marrec for help with figures, and Amanda Montalbano for excellent logistical support.

### Funding

This work was conducted within the NASA supported North Atlantic Aerosols and Marine Ecosystems Study (NAAMES, grant NNX15AL2G). Further support for data analysis and synthesis was provided by the NASA EXport Processes in the global Ocean from RemoTe Sensing (EXPORTS) field campaign (grant 80NSSC17K0716 to Susanne Menden Deuer and Tatiana Rynearson). There was no additional external funding received for this study. The funders had no role in study design, data collection and analysis, decision to publish, or preparation of the manuscript.

### Grant Disclosures

The following grant information was disclosed by the authors:
NASA supported North Atlantic Aerosols and Marine Ecosystems Study (NAAMES): NNX15AL2G.
NASA EXport Processes in the global Ocean from RemoTe Sensing (EXPORTS): 80NSSC17K0716.

### Competing Interests

Susanne Menden Deuer is an Academic Editor for PeerJ. The other authors declare they have no competing interests.

## Author Contributions

- Francoise Morison, Andreas Oikonomou conceived and designed the experiments, performed the experiments, analyzed the data, prepared figures and/or tables, authored or reviewed drafts of the paper, and approved the final draft.
- James Joseph Pierson analyzed the data, authored or reviewed drafts of the paper, and approved the final draft.
- Susanne Menden-Deuer conceived and designed the experiments, performed the experiments, authored or reviewed drafts of the paper, and approved the final draft.

## Data Availability

The data generated for this study are available at the SeaWiFS Bio-optical Archive and Storage System (SeaBASS) maintained by the NASA Ocean Biology Processing Group.

https://seabass.gsfc.nasa.gov/archive/URI/menden-deuer/NAAMES/naames_2/archive

The files from the archive specific to the present study are:

- NAAMES2_AT34_MendenDeuer_Mesozoo_S1.sb
- NAAMES2_AT34_MendenDeuer_Mesozoo_S2.sb
- NAAMES2_AT34_MendenDeuer_Mesozoo_S4.sb.

## Supplemental Information

Supplemental information for this article can be found online at http://dx.doi.org/10.7717/peerj.9430#supplemental-information.

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
