# Peer review of "Mesozooplankton grazing minimally impacts phytoplankton abundance during spring in the western North Atlantic"

_PeerJ, doi:10.7717/peerj.9430_

## Round 0.1 · original submission · Major Revisions

While the referees are positive about the paper they make extensive suggestions for improvements. Please attend to these carefully and systematically, and do another good proofread before resubmitting. The Discussion must be clear on the study limitations, particularly the very artificial nature of the experiments.

·

Basic reporting

The study is generally well performed and the ms well structured and written by native English speakers (as presumed by names and affiliations) in clear, professional language. As a non-native English speaker I cannot comment further on that. The Introduction to the ms provides adequate background information and puts the problem under investigation in a proper reference framework; important and relevant bibliographic references are provided in support of authors’ statements. There are, however, a few instances where the message may not be completely clear. For instance:
I found a bit confusing the following statement (lines 64 – 66): “Thus microzooplankton are thought to respond faster than mesozooplankton to increasing phytoplankton availability leading to bloom events (Sherr & Sherr 2009).”. The first part of the sentence is clear-cut, but I do not see how the second part is logically derived from the first. I mean that microzooplankton grazing is usually considered a first order regulator of phytoplankton population-community dynamics, preventing the formation of blooms on most occasions (i.e., blooms happen when grazing control fails; for instance Irigoien, Flynn & Harris 2005, JPR 27: 313-321, Beherenfeldt 2010, Ecology 91: 977–989; Beherenfeldt & Boss 2018, Glob Change Biol 24: 55–77; ). As it stands, the second part of the sentence seems contradictory with such understanding. Could authors please rephrase?
Also, the following statement seems unclear to me, or not completely correct (lines 66 – 68) “While grazing pressure of mesozooplankton on oceanic primary production (PP) can occasionally be comparable to that of microzooplankton, in productive oceanic regions microzooplankton tend to exert a higher grazing pressure than mesozooplankton”. Microzooplankton grazing rates tend to be higher than mesozoo’s on both high and low productive ecosystems, according to the series of meta-analyses and synthesis by Calbet and collaborators (Calbet & Landry 2004, Limnol. Oceanogr., 49: 51–57; Calbet 2001, Limnol. Oceanogr., 46: 1824–1830; Calbet 2008, ICES J Mar Sci 65: 325– 331; Calbet & Saiz 2005, Aquat Microbial Ecol 38: 157– 167).

Figures are all relevant and clearly presented. One suggestion would be to provide an extra figure (maybe as supplementary matter) to show the results from the dilution experiments, i.e., net growth vs %WSW. That would aid readers to follow better some of the deductions and interpretations of results from Sts 2 & 4. No reference is made to raw data availability.

Experimental design

A general problem I find in relation to the approach to derive grazing estimates is that when both meso and microzooplankton are present in the bottles, grazing rates by both grazer communities are not additive due to interactions among the many components, including intra-guild predation. Strictly speaking that is a problem inherent to the seawater dilution method where complex micro to pico-sized communities are manipulated. Adding mesozooplankton enhances the chances and ranges of such interactions, and thus increases the difficulties in the interpretation of the experimental designs. In the Discussion section authors properly address some of this issues.

Besides the above, the methods followed are currently largely accepted. I highlight positive issues like the correction performed to take into account photo-acclimation effects. Indeed, the validity of the whole approach based on chlorophyll would be seriously compromised without that improvement.

It is unclear the reason why C. hyperboreus was selected as representative copepod of S1 considering that the most abundant was C. finmarchicus. Could authods justify their choice?.

The description of the methods is generally fine, but there are several aspects that are not well covered, or simply deserve further detail. Main issues follow:
- filtered (S1) or unfiltered (S2) seawater was used to keep zooplankton for later sorting. why not a consistent protocol was followed?. Please justify.

- Use of a 220 um mesh net clearly underestimates abundance of small copepod species (Paracalanus spp. and Oithona spp.) and also naupliar stages of larger ones (Calanus spp.). Those issues could be somehow accounted for when discussing the field implications of current results.

-How much unscreened water was added to the experimental bottles (copepods treatment) at S4?. Did that resulted in approximately realistic copepods concentrations within the bottles?.

- It is not stated in the methods whether different zooplankton concentration treatments were also considered at S4?. Please specify.

-Please indicate the volume of water analysed in order to characterize particles size spectra.

Validity of the findings

No comment

Additional comments

Other issues:

Results
It is stated that “No microzooplankton grazing was detected at S2” (lines 292-293). Could authors please be more specific?. One understands that net growth measured at 100%WSW was exactly the same than that measured at 20%WSW. If actual results were shown this doubt would be easily evacuated.

It is important that the number (or concentration) and relative composition of the mesozooplankton in the experimental bottles is also reported (for S4). A priori one may expect relative composition inside the bottles to be identical to that in the field community, but due to all necessary manipulations for the preparation and addition of the sample to the bottles, the experimental community may deviate significantly from the field one.
Related to the above: In line 300 text reads “Inclusions of mesozooplankton at in situ abundances and… “. Which indications exist that mesozoo concentration inside the bottles properly represented in situ abundance?. Could authors present evidence on that?

I assume particles size spectra shown in Fig. 6 are averages from replicate samples/ bottles. Please specify, and provide some error index for between-replicates variability (could be just a quote in the text, or in the legend to thefigure).

It is stated that “… all particles in the size fraction 15-60 μm were removed by mesozooplankton grazers at all stations over the course of the 24 h incubations” (lines 338-340). However, Fig. 6 clearly shows the presence of particles >15 um in all stations and under all conditions tested. Maybe I am not properly interpreting these results; could authors revise and/ or explain better?.

Lines 348-350: “Blank controls contained a negligible amount of particles and replicate samples revealed a coefficient of variation of 1%.”. Please specify if that CV refers to the total particle count from replicate FSW samples.

Discussion

At St. 4 particles abundances seem to be slightly higher in Microzoo treatment compared to the initial condition. As the ordinates axis is log10-scaled, at certain size classes the difference may be substantial. Any clue which may be the reason for that?. What about population growth?

I do not completely agree with the statements in lines 366-369. And consequently with the proposed shifting balance between micro and mesozoo grazing. Justification: Grazing by microzoo at St 4 was undetermined, not zero. Those results are obscured by the fact that in that case no photo-acclimation correction could be made. The unexpected pattern (lower growth at 20%WSW) could be related to that. In any case, if the basic assumptions of the method were not met, that does not necessarily leads to the conclusion that grazing was zero. This issue is revisited later in the Discussion (line 474), and this second time the interpretation is different and more rigorous (and correct, in my opinion).

Lines 422-423: “The low impact of grazing by copepods on phytoplankton observed here suggests that copepods likely relied on other food sources.” That is not necessarily so. Copepods may obtain 100% of their required daily ration from phytoplankton and still use a small fraction of primary production. It is a matter of copepods field biomass concentration and biomass-specific energetic requirements: copepods are most of the time too few per unit ocean volume to induce a higher impact on primary production (please note: indirectly this is correctly discussed on previous paragraphs that deal with potential aggregation effects). Of course, on top of that, copepods also consume other non-phytoplankton items. We completely agree on that. Please rephrase.

Consider that copepod densities used in the bottles of S2 are very likely unrealistically high (10000 – 40000 C. finmarchicus m-3). That may have helped to get a clearer signal but at the same time may have induced an underestimation of activity (grazing) rates. Probably a similar consideration is valid for C. hyperboreus at S1?.

Statement in lines 437-439: Yes, I generally agree with the speculation that copepods likely preyed upon both microzoo and phytoplankton to some extent. But consider that meta analyses suggest that there is an overall trend for copepods to select microzoo (actually ciliates) over phytoplankton (Calbet & Saiz, op. cit).
448-449: statement “… the situation would be similar in situ, as feeding on larger cells would be limited by the low in situ abundance of larger particles”. Well, not necessarily so. Limitation is more likely during the experiments due to very high copepods concentrations used and limited bottle volume.

Lines 495-516 (par on light effects): General agreement. But, for the lack of light effects on grazing consider the following: if copepods grazing is developed mostly during night ours and only marginally during daytime, then light intensity during non-feeding hours may irrelevant and not lead to a light effect on grazing rates (under the assumption than the photoperiod remains constant, as it is the case here).

Reviewer 2 ·

Basic reporting

see summary text

Experimental design

see summary text

Validity of the findings

see summary text

Additional comments

This manuscript present results from three sampling in the North Atlantic during the spring bloom. The stations are widely separated and range from 4-15 °C, and obviously two very different water masses. S1 and 2 are arctic with large Calanus dominant, S4 är temperate with a dominance of smaller copepods. At each station the microzooplankton and mesozooplankton herbivorous grazing is measured or inferred through the dilution technique. The particle size spectrum is measured at the three stations, before and after incubation with copepods.

The experiments are OK and well presented, but only 2-3 replicates are set up. The addition of copepods is very high, 5 C. hyperboreus and up to 40 C. finmarchicus added to 1.2 L bottles is very crowded. Hence clearance rates are very low for both species. Certainly, they do not perform anywhere near natural under these conditions. Therefore, any attempt to measure grazing potential (either as %PP or % phytoplankton biomass) will likely be underestimated. Indeed, the authors discuss this at ambient abundances and conclude that the grazing should be dominated by microzooplankton at S1. This should probably be so at S2 and S4, but here they did not get a negative slope and thus could not estimate the microzooplankton grazing.

At S2 and S4 there should be strong cascading effects of copepods feeding on the microzooplankton. The authors should take this into account. C. hyperboreus probably do not feed on ciliates, but C. finmarchicus and the smaller copepod assemblage would certainly change the ciliate concentrations when added at these high abundances. It would be interesting to note this in the analyses and in the discussion.

The chlorophyll was analysed by extraction, but indicate what pore size filters were used. This important since the flow cytometer also collect particles smaller than 0.7 µm, the standard nominal pore size of GF/F filters. The dilution series were prepared by 0.45 µm screened water, were these filters also used for the chla determinations?

The grazing is inferred from the slope in Fig 4, but the regression tests are wrong. The df in line 302 and 307 regressions is 1, not 6. Significance seems OK.

The two-way ANOVA should be presented with all relevant df, SS, MS, F and p-values. The discussion of the results from the ANOVA (lines 317-333) is unclear, but will be clearer if the table is presented.

What was the absolute abundance of the copepods? (Fig 3). Please give this along with percent composition.

Fig 6. Are there any replicates for these data? Error bars for the measures? You should also provide statistical tests for the differences between treatments, e.g. Kolmogorov-Smirnov tests

Table 1. The ABC for the different bottles are not shown as is indicated in the legend, please correct

Line 530 This conclusion (no preferential feeding …) is not well supported by data. Explain better or remove.

·

Basic reporting

The manuscript reads very well and generally speaking the flow and contextualization is great. The objectives are clearly stated. It is very nice to see studies on micro-zooplankton grazing rates combined with meso-zooplankton, as this is often ignored and the former, as shown here, can be quite significant. However, because the micro-zooplankton grazing measurement is often ignored, many readers will not be familiar with the dilution method, even less so with the modified method used here. Thus, below are some general and specific suggestions to improve the flow and make it easier for readers to understand and replicate the study.

General issues

1) I would suggest a graphical representation to visualize the experimental design for the grazing experiments. The method is quite complex with a specific concentration gradient of meso-zooplankton at each site crossed with different nutrient conditions, micro-zooplankton grazing via dilution experiments, and also different light intensity in one of the sites (S4). Thus, it may be difficult for many readers to understand how the micro- and meso-zooplankton grazing effect was calculated and compared, especially for those that are not familiar with the modified dilution method. I suggest including a graphical image of the overall experimental design to measure the micro- and meso-zooplankton grazing for S1, S2 and S4. This would undoubtedly help readers understand and provide a clear vision for anyone who wishes to replicate or build on what was done.

2) The grazer effects and comparison of the differences among micro- and meso-zooplankton effects presented in Table 1 (or from the other results) is not straightforward. It might help to calculate the overall effect of each grazer type on % primary production, for example (see more on this below in “validity of findings”).

3) I suggest to highlight more clearly in the results that S1 had the most grazing by micro-zooplankton, that S2 had the largest impact of copepods on particle size distribution but no observable microzooplankton grazing, and that S4 had no significant grazing effect by either meso-zooplankton or micro-zooplankton.

Specific issues:

L 241-242: Many readers will not be too familiar with the modified method for calculating micro-zooplankton grazing rates used here. In this context, the difference between k (net phytoplankton growth rate) and µ (in-situ specific phytoplankton growth rate) and their importance for the method can be better explained. Moreover, I suggest calling µ the “total phytoplankton growth rate”, which is more descriptive than “specific growth” (if I understood correctly, µ is the total growth rate = grazing + k). This also goes for Table 1.

L 242: “the sum of the grazing rate and the mean net growth rate of the non-amended <200 μm WSW bottles”. I understand that the grazing rate here is the micro-zooplankton grazing rate (i.e., “g”). If so, please make this clear by adding a parenthesis “(g)” after “grazing rate”. Also, why is the µ in Table 1 not the sum of (g + k NoN)? If I understand correctly, mean net growth rate of the non-amended <200 μm WSW bottles is k NoN in Table 1. Thus, µ should be the sum of g + k NoN (=0.40, not 0.39), right?

L 346-348: Do you mean for all sites? If so, please state clearly.

L 361: “… observed shifting balances …” shifting in what context? “Shifting balances” here may be replaced with “We observed significant variability in the micro- and mesozooplankton grazing impact” to avoid the possible temporal implications that may be associated to “shifting”.

L 368: Why is this remarkable? Would light limitation not be expected to be a major loss factor here? Please explain in a few words.

Table 1: I believe A, B, and C are missing in the column for k Cop.

Figures 4 and 6: There is data for all of the treatment copepod concentrations in Figure 4 for S1 and S2 (i.e., 3 per site) but not in Figure 6? Please add the data for these different copepod abundances or explain why they are not included.

Experimental design

The topic is within the scope of journal and there is a well defined research question. The authors did a good job in describing the relevance of this work and highlighting the knowledge gaps. The methods are sound but could be explained in a more straightforward fashion, especially with regard to why the nutrient additions were made and the logic for using nutrient added or non-added bottles while calculating grazing impacts. Doing so would improve future efforts in replicating the methods and approach used here. Please find below some suggestions for improving the manuscript in this context.

General comments

1) It is not straightforward why the micro-zooplankton grazing rates are based on phytoplankton growth rates from the bottles with nutrient additions while the net phytoplankton growth is calculated from non-amended bottles. Please explain the reason for this and see the specific comments below. This would also help to understand and interpret results presented in Table 1.

2) Given the difference in copepod size in S1 and S2, it would be nice to provide the length and biomass of the species used. That way the reader and the authors can compare the mass-specific top down effects.

3) The in-situ grazer density (i.e., natural density during sampling) is not reported, which makes it difficult to compare the experimental grazer density with that of the natural grazer density at the time of the experiment. The authors mention that it was not possible to estimate this density but it would great to include some estimates from other studies to provide the range that might be expected in nature.

Specific comments

L 235: Why is the microzooplankton grazing rate (g) calculated from phytoplankton growth rates (k) obtained from the bottles with nutrient addition only (k +N)? I believe many readers, including myself (!) would like to know the justification for this.

L 242: Similar to the comment above for L 235, please explain why µ is calculated using the net phytoplankton growth from non-amended bottles? It is interesting that g is calculated from k + N, while µ is calculated from k NoN and the conceptual reason for this should be clear.

L 341-345: What happened to the data from the bottles with different copepod densities for S1 and S2? Please explain why there are 3 different copepod concentrations in the grazing experiment (S1 and S2) but only 1 copepod concentration for S1 and 2 copepod concentrations for S2 in Figure 6?

Validity of the findings

Overall, the conclusions and their relevance to the objectives are clearly stated and are mostly justified by the results presented. Combining no-grazer controls in addition to different grazer concentrations adds to the strength of the meso-zooplankton grazing experiments. One key limitation might be the lack of a universal metric to compare the effect of micro- versus meso-zooplankton on phytoplankton growth (e.g., % reduction in PP). There are a few more suggestions for improving the manuscript in terms of strengthening the conclusions provided below.

1) Currently, the relative effect of micro-zooplankton is not directly compared to the effect of meso-zooplankton on phytoplankton. Thus, the justification for concluding that micro-zooplankton had a stronger effect compared to meso-zoopankton is not straightforward. I suggest the authors to make this comparison clearly before proposing general conclusions between the grazing impact of micro- vs. meso-zooplankton. For example at S1, did microzooplankton graze more phytoplankton or did copepods? L 292 says that 97% of primary production was grazed by micro-zooplankton, but this is for the dilution experiment only. How does this compare to what was grazed down by copepods at the same site (S1)? Based on the k copepod values in Table 1, can we conclude that the highest copepod density in S1 (k Cop= - 0.40 relative to the k NoN = 0.02; a difference of 0.42 d-1) consumed more phytoplankton than micro-zooplankton (g = 0.38)? It is not clear if this kind of comparison is possible. Either way, the quantitative effect (e.g., % reduction in phytoplankton biomass or growth) among the micro- vs. meso-zooplankton should be made clear for S1 and also estimated for S2 and S4. If this is not possible, please explain why not.

2) The conclusions regarding the grazing effect of copepods on micro-zooplankton (e.g., Line 530) may not be entirely justified because there is no specific information on this in the results. Please either re-formulate these conclusions, or clearly state in the results what data there is showing the grazing of copepods on micro-zooplankton. More on this in the specific comments below.

3) It would add significantly if the authors could include the mass-specific CR data (i.e., mg prey mg copepod-1 day-1). The prey here could be in terms of chl-a, or biovolume. This would enable others to compare among previously published grazing rates and give an idea of the grazing pressure to those more familiar with CR. See specific comments below for more on this.

4) It is very nice how the authors showed the effect of copepods on the prey size spectrum. Would it be possible to estimate size specific grazing rates? If the copepods ingested large sized phytoplankton, perhaps it would be interesting to estimate the CR based on small vs. large prey.

5) Given the latitudinal gradient of the study, it might be interesting to speculate about the geographical differences in the observed results. While the opportunistic sampling indeed limits such generalizations (as recognized by the authors), it is interesting that the colder, more mixed water (typically where we might expect stronger copepod grazing pressure on phytoplankton) had the strongest micro-zooplankton grazing impact. What do the authors think about this? Just a result of stochastic mechanisms? A few words on this in the conclusion would add to the manuscript.

6) Would meso-zooplankton grazers including copepods increase at night due to vertical migration? By what factor would this increase the grazer abundance? Would be interesting provide a few lines on the possible shifts in copepod grazing pressure due to vertical migration.

Specific comments

L 249: It is difficult to compare per capita CR among different sized grazers. Larger grazers will have a higher CR. Thus, it is more appropriate and informative to compare mass specific CR. That way one may compare the top down effect in different sites and other workers, including models, may compare with your results. If possible, please provide the mass-specific CR estimated from S1 and S2 for the different copepod species used.

L 317: Is it not possible to also calculate the mass-specific CR from S4? If the number and size of the copepods and other meso-zooplankton in the grazing bottle is known, than it would be possible and useful to calculate mass-specific CR. By doing this also for the S1 and S2 experiments, then you could compare the mass-specific CR across all sites for a more integrative view on mesozooplankton grazing. I believe this would make the study stronger and more informative.

L 361-363: Please see the general comment #1 under the Validity section.

L 366-368: Could the % reduction of phytoplankton growth due to meso-zooplankton be estimated for S2 and S4?

L 384-395: This is great, but please also add some published copepod densities reported for this region to back up the statements made. For example, what is the common recorded copepod abundance and the exceptionally high “peak” abundances in the NW or mid-Atlantic? These numbers would make the point much stronger and add to the empirical context of the presented results.

L 419-421: Having a clearly stated effect of copepods on PP (% reduction) from this study would help put these results in context.

L 452-454: Unfortunately, this statement, as it is currently written, is not justified by the results. If I understand correctly, there is no information in the current study regarding the grazing impact of copepods on micro-zooplankton. Also, is the 15% reduction in PP a result from this study? If so, from which site? Please make this information more straightforward.

L 454: Which size class exactly? Please specify.

L 530: Please indicate where in the results this information about the degree of micro-zooplankton grazed by copepods is presented.

---

## Round 0.2 · Minor Revisions

Thank you for the carefully revised MS. One referee makes some good suggestions for more details in the methods. I also suggest avoid starting sentences with abbreviations (e.g., C. helgolandicus) and consistently rounding % to whole % in the text because it makes for easier reading. My compliments of the figures being nice and clear. [So many papers have microscopic text and poor colour combinations (even in top journals). ]

·

Basic reporting

No comments

Experimental design

No further comments

Validity of the findings

No comments:

Additional comments

Answers to comments on the first round of review show an effort to consider all issues raised by different reviewers. That means inclusion of extra information required (for instance regarding the methods) as well as taking care of interpretation issues or clarifications requested. I find the ms is in better shape now.
A single further comment (I must have overlooked this on the first round) is that I could not find a reference in the Methods to actual dates when stations were occupied and experiments run. At the end of the Introduction it is stated that this investigation was carried out during the NAAMES spring campaign. It is good to mention that. However, oceanographic condition during for example end of March may be rather different from that in early June, and both correspond to Spring. Similarly, information on the year the oceanographic cruise was set is in order, as year to year environmental variability could be significant. I think provision of more precise information on the actual year and dates is valuable for reference and comparison for future investigations, and will thus enhance the utility of current data and results. I thus strongly recommend the addition of that information.

·

Basic reporting

The authors did a good job at revising their manuscript and I have no further comments or suggestions.

Experimental design

The authors did a good job at revising their manuscript and I have no further comments or suggestions.

Validity of the findings

The authors did a good job at revising their manuscript and I have no further comments or suggestions.

---

## Round 0.3 · accepted · Accept

Thanks for addressing those small details and publishing with PeerJ.